# Heterogeneous Ensemble Deep Learning Model for Enhanced Arabic Sentiment Analysis

**DOI:** 10.3390/s22103707

**Published:** 2022-05-12

**Authors:** Hager Saleh, Sherif Mostafa, Abdullah Alharbi, Shaker El-Sappagh, Tamim Alkhalifah

**Affiliations:** 1Faculty of Computers and Artificial Intelligence, South Valley University, Hurghada 84511, Egypt; sherif.mostafa@fcih.svu.edu.eg; 2Department of Information Technology, College of Computers and Information Technology, Taif University, P.O. Box 11099, Taif 21944, Saudi Arabia; amharbi@tu.edu.sa; 3Faculty of Computer Science and Engineering, Galala University, Suez 435611, Egypt; sh.elsappagh@gmail.com; 4Information Systems Department, Faculty of Computers and Artificial Intelligence, Benha University, Banha 13518, Egypt; 5Department of Computer, College of Science and Arts in Ar Rass, Qassim University, Buraydah 52571, Saudi Arabia

**Keywords:** machine learning, deep learning, ensemble learning, Arabic sentiment analysis

## Abstract

Sentiment analysis was nominated as a hot research topic a decade ago for its increasing importance in analyzing the people’s opinions extracted from social media platforms. Although the Arabic language has a significant share of the content shared across social media platforms, its content’s sentiment analysis is still limited due to its complex morphological structures and the varieties of dialects. Traditional machine learning and deep neural algorithms have been used in a variety of studies to predict sentiment analysis. Therefore, a need of changing current mechanisms is required to increase the accuracy of sentiment analysis prediction. This paper proposed an optimized heterogeneous stacking ensemble model for enhancing the performance of Arabic sentiment analysis. The proposed model combines three different of pre-trained Deep Learning (DL) models: Recurrent Neural Network (RNN), Long Short-Term Memory (LSTM), Gated Recurrent Unit (GRU) in conjunction with three meta-learners Logistic Regression (LR), Random Forest (RF), and Support Vector Machine (SVM) in order to enhance model’s performance for predicting Arabic sentiment analysis. The performance of the proposed model with RNN, LSTM, GRU, and the five regular ML techniques: Decision Tree (DT), LR, K-Nearest Neighbor (KNN), RF, and Naive Bayes (NB) are compared using three benchmarks Arabic dataset. Parameters of Machine Learning (ML) and DL are optimized using Grid search and KerasTuner, respectively. Accuracy, precision, recall, and f1-score were applied to evaluate the performance of the models and validate the results. The results show that the proposed ensemble model has achieved the best performance for each dataset compared with other models.

## 1. Introduction

With the noticeable increase and availability of internet forums, blogs, press sites, and social networks, people have the opportunity to show and express their sentiments and opinions publicly available to everyone. The steady increase in information and data volumes created a new branch of science called sentiment analysis (SA). Sentiment analysis can be summarized as the operation of analyzing opinions and also emotions to deduce the tendencies appearing in the analyzed data, classifying them as positive, negative, or even neutral. Sentiment analysis helps companies realize people’s opinions on different topics and even various commodities, which has a tangible impact on helping the concerned companies make the right economic and productivity decisions at the right time. Its importance is extended even to the financial markets and stock exchange [1].

The importance of the Arabic language and its position, which the dominant languages may parallel in the world, came from the that Arabic is one of the six official languages used within the united nations and the mother tongue for about 300 million people, occupying 22 different countries. Modern Standard Arabic (MSA) forms the formal language for communication, which most Arabic-speaking people can understand [2]. MSA is commonly used in radio, newspapers, and television. All these reasons lead to the inflation and increase of Arab markets, and the expanding usage of media platforms leads to the interest in analyzing data in Arabic despite its multiplicity of dialects [3].

ML algorithms utilize diverse statistical, optimization, and probabilistic calculations and algorithms to absorb and comprehend experience to sketch functional patterns from unstructured, large, and complex datasets [4]. These algorithms could be used in a variety of applications that have an impact on the public and their daily lives, such as automated text categorization [5], network intrusion detection [6], customer purchase behavior detection [7], disease modeling [8].

Recently, the research used ML algorithms to predict sentiment analysis and to enhance the sentiment analysis method’s accuracy [9,10,11], transit from primarily linear models and evolved until it reached what we see today of the more complex deep neural network. In addition, deep learning (DL) algorithms have been used to extract features with great effectiveness over other ML algorithms such as [12,13].

Ensemble learning has recently played a vital role in natural language processing to enhance results. The essential objective behind an ensemble of models is to aggregate several base classifiers so that their combined performance compensates for and outperforms the accuracy of each model function individually. Two points have a great impact on ensemble learning performance, how predictions and base learners are combined, is meta-learning or rule-based; also, the methods followed to complete the learning process, either it’s sequential or concurrent [14,15]. Heterogeneous ensembles are constituted of classifiers of diverse types, whereas homogeneous ensembles are made up of classifiers of the same type. The strategies used to produce variation among the basis classifiers for homogeneous and heterogeneous ensembles are different. Heterogeneous ensembles composed of predictors represent different types, possibly having different biases. The combination of biased decisions could be superior to homogeneous ensembles. If these biases are complementary [16]. In most cases, Ensemble learning methods can be in the form of three popular ones, namely bagging [17], boosting [18], and stacking [19]. Many Researcher activities on ensemble learning centered around homogeneous ensembles, even though heterogeneous ensembles could prove more efficient in case of combining pre-trained models that are often readily available such as [20,21].

### Motivation and Contribution

The challenges of this era have made data and information analysis one of the most vital things. Its importance comes from data sentiment is the way to obtain valuable information that can be used and relied upon. As for its vitality, it appears that errors in data analysis or interpretation will lead to many problems for the applications that took that data as their input or can lead to wrong predictions resulting in dangerous consequences. From here, it was necessary to have an elaborate and decisive way of interpreting information, especially information from languages other than English, such as Arabic. As a result of the increasing enthusiasm for infusing the Arabic language into various disciplines related to computing, there has become a significant volume of Arabic language data. In addition, ensemble methods could increase computational cost and complexity due to the expertise and time required to train and maintain multiple models rather than a single model [22] However, this point can be overlooked in front of the advantages that could be gained when applying this method. There are two main reasons to use an ensemble over a single model. First, ensemble techniques always achieve better performance than all of its baseline single learner [23]. This gain in performance can be explained as the model introducing bias to reduce the variance component of the prediction error [24]. Second, Ensemble reduces the spread or dispersion of the predictions, resulting in improved model robustness, stability, and reliability in the average performance of a model [24]. In more general terms, Ensemble is the optimal solution with the best situations when compared to individual learner performance in many problems and situations [25].

A stacking ensemble model has been proposed in this paper that combines three different DL models, Recurrent Neural Network (RNN), Long Short-Term Memory (LSTM), Gated Recurrent Unit (GRU), using three meta-learners, Logistic Regression (LR), Random Forest (RF), and Support Vector Machine (SVM), to enhance the model’s performance for predicting Arabic sentiment analysis. We also compare the performance of the proposed model with RNN, LSTM, GRU, and five regular ML models: Decision Tree (DT), LR, K-nearest Neighbor (KNN), RF, and Naive Bayes (NB), using three benchmark datasets.

The contributions of this study are as follows:We proposed a stacking ensemble model that combines DL models (RNN, LSTM, and GRU) using three meta-learners (LR, RF, and SVM) to enhance performance models of predicting Arabic sentiment analysis.We optimize ML models, DL models, and proposed models using optimization methods to enhance performance.We conduct various experiments to evaluate the performance of the proposed model using three benchmark datasets.We compare the proposed model’s performance with different ML and DL algorithms.

The remainder of the paper is organized as follows: Section 2 reviews related works on sentiment analysis. The proposed model is presented in Section 3. The experimental results and discussion are presented in Section 4. Finally, Section 6 provides a summary of the paper.

## 2. Related Work

Research have been used ML, DL models to predict sentiment analysis. For example, The authors in [26] proposed a hybrid model that integrates convolutional neural network CNNs and LSTMs for predicting sentiment analysis. They used Arabic Health Services Dataset (Main-AHS and Sub-AHS), ASTD, and Ar-Twitter. The study concluded that the 5-gram with CNN-LSTM has better sentiment classification results. Researchers in [27] proposed an efficient Bidirectional LSTM Network (BiLSTM) coupled with the ability of extraction features. The information extracted from the feature sequences is based on both the forward and backward dependencies. Experiments were conducted using six benchmark sentiment analysis datasets to evaluate the performance of models. The results show that the proposed model achieved significant improvements over the other models: SVM, RF, and LSTM. In [28], authors investigated various DL models based on CNN and LSTM for sentiment analysis of Arabic microblogs. They conducted experiments using CBOW, skip-gram (SG), and ASTD and Ar-Twitter datasets. The experiments showed that LSTM performs better than CNN. The authors [29] applied SVM and NB with different weighting schemes (TF, TF-IDF) and n-gram sizes to predict Arabic sentiment analysis. They conducted experiments using the AJGT dataset and the SVM classifier’s best-performing scenario. In [30], the authors proposed the hybrid models-based DL algorithms for sentiment classification. They used more than 1 million tweets in different domains and compared the hybrid model with RF, DT, RNN-LSTM, CNN, hybrid models. The result showed the hybrid model has the best performance. A. M. Alayba et al. [31] used different ML models: NB, SVM, LR Stochastic Gradient Descent (SGD), Ridge Classifier (RDG), and DL model: CNN with other feature extraction methods to predict Arabic sentiment analysis. Experiments were conducted using Arabic Health Services Dataset (Main-AHS and Sub-AHS). Mohamed Fawzy et al. [32] discussed a variety of DL network architectures used for Arabic sentiment classification coupled along with the word embedding approaches. RNN, CNN, Bidirectional Multi-Layer LSTM with different word embedding to do experiments. Experiments were conducted using Large-scale Arabic book reviews (LABR). The result showed that Bidirectional Multi-Layer LSTM has high accuracy. In [12], author applied SVM, LR, DT, NB, and DL models on the Saudi dialect sentiment Arabic tweets dataset. The result shows that deep learning and SVM classifiers perform best with accuracy.

Some research used ensemble learning approach to predict sentiment analysis. For example, Al-Hashedi et al. [33] used NB, SGD, RF, LR, and voting classifier is an ensemble classification method to predict sentiment analysis. The author collected Arabic tweets about COVID-19 and annotated tweets into positive and negative. The result shows that the voting classifier has high performance. Alharbi et al. [34] proposed a DeepASA model consisting of an input layer, the hidden layers, two types of DL networks were used GRU and LSTM, and the final layer was used a voting system boost the model’s prediction performance. Experiments were conducted using different Arabic datasets: Large Scale Arabic Book Reviews Dataset (LABR), Hotel Reviews (HTL), Restaurant Reviews (RES), Product Reviews (PROD), ArTwitter, and ASTD datasets. The result shows that the DeepASA model has high performance. Oussous et al. [35] applied a voting algorithm on top of three classifiers, SVM, NB, and Maximum Entropy, on the ASTD Arabic sentiment analysis dataset. The results show vote algorithm has high accuracy. Al-Saqqa et al. [36] proposed an ensemble of four ML classifiers, KNN, SVM, NB, and based on the majority voting algorithm to classify the sentiment of Arabic text. Three varied size datasets were used: movie reviews, ArTwitter, and large-scale Arabic sentiment analysis dataset (LABR). The experiments revealed that the ensemble of the classifiers gives better results than individual classifiers. Al-Azani et al. [37] compared the performance of different ensemble learning techniques to boost the performance of individual classifiers, including bagging, boosting, voting, stacking, and RF on the Arabic sentiment dataset. The result shows that the stacking ensemble has high performance.

Other authors applied ensemble learning techniques for sentiment analysis with non-Arabic languages. For example, Sitaula et al. [38] prepared a Nepali Twitter sentiment dataset called the NepCOV19Tweets, and labeled it positive, neutral, and negative. The authors proposed feature extraction methods using different feature selection techniques including fastText, domain-specific, and domain-agnostic. They used different CNN models to implement each feature selection method. Then, they proposed a CNN ensemble model to capture multi-scale information for better classification. In [39], the authors proposed a multi-channel CNNs (MCNN) to classify the NepCOV19Tweets dataset into positive, neutral, and negative sentiment classifications. Their proposed MCNN model was trained using a hybrid feature extraction method for semantic and syntactic features. The proposed hybrid features achieved the highest accuracy compared to individual feature extraction methods, and the MCNN model had the highest accuracy.

Previous studies used regular ML, hybrid models, and homogeneous ensemble learning. However, they did not use heterogeneous ensemble learning. In our study, we proposed heterogeneous ensemble deep learning model for enhanced Arabic sentiment analysis.The proposed model combined three different DL models including RNN, LSTM, and GRU. We explored three meta-learners including LR, RF, and SVM to enhance the model’s performance for predicting Arabic sentiment analysis.

## 3. Methodology

The framework for predicting sentiment analysis for Arabic data includes the ML, DL, and ensemble learning approaches as shown in Figure 1. In the ML approach, five regular ML models are used: NB, KNN, DT, RF, and LR. Term frequency-inverse document frequency (TF-IDF) with different sizes n-gram is used as a feature extraction method, and grid search with cross-validation is used to optimize ML models. In the DL approach, three models: RNN, LSTM, and GRU, are used. CBOW word embedding is used as the feature extraction method. The Keras-tuner is used to optimize the DL models. In ensemble learning, we proposed the stacking ensemble model that combines RNN, LSTM, and GRU that are developed in the second approach using three meta-learners: LR, RF, and SVM. The grid search is used to optimize meta-learners. Each approach will be described in detail.

### 3.1. Data Collection

Three benchmarks of Arabic sentiment analysis datasets are used in this paper.

#### Dataset1: Arabic Sentiment Twitter Corpus (ASTC)

Arabic Sentiment Twitter Corpus (ASTC) [40] was collected in April 2019 from Twitter. In total, there are 56,795 Arabic tweets with positive and negative labels. It is divided into two parts: a training set and a test set.There are 22,626 negative classes and 22,810 positive classes in training set. There are 5703 positive and 5656 negative classes in the testing set.

### 3.2. Dataset2: (ArTwitter)

ArTwitter [41] was collected from Twitter on different topics: politics and arts. It consists of 1951 Arabic tweets annotated in positive and negative labels. It is split into 80% training set and 20% testing set. The training set contains 794 positive and 766 negative classes. The testing set contains 199 positive and 192 negative classes.

#### Dataset3 (AJGT)

Arabic Jordanian General Tweets (AJGT) [42] dataset consists of 1800 tweets have been annotated as either positive or negative. It is divided into a training set of 80% and a testing set of 20%. A total of 720 positive and negative classes are included in the training set. A total of 180 classes are classified as positive and “negative” in the testing set.

### 3.3. Data Pre-Processing

Because the text of tweets is known to be noisy, it must be cleaned and pre-processed before being analyzed.

Cleaning Tweets: The removal of irrelevant information is crucial to cleaning Twitter data due to its noisy nature. By removing non-Arabic letters, digits, single Arabic letters, and special symbols, removing URLs, removing Emails, removing hashtags.Tokenizing: Tokenization only involves segmenting the sentences into parts. Tweets will be tokenized by splitting text by space.Removing Arabic Stop Words: The data processing will be more efficient by excluding Arabic stop words. Some of the stop words are removed from the original tweets.Stemming: The stemmer’s main job is to return the word construction to the base word (root or stem). One Arabic stemmer included in the NLTK package is ISRI Stemmer [43].Cleaning Emoticons: Twitter users employ symbols such as “: D” and “;)” to communicate their feelings and opinions. These emoticons, also known as emojis, convey important information. As a result, they are labeled to discern the sentiment behind the emotions. Therefore, we remove emojis, translate emojis, and conduct emoji Unicode translation.

### 3.4. Splitting Dataset

Each dataset was separated into two sets: the training set and the testing set, with the training set accounting for 80% of the dataset and the testing set accounting for 20% of the dataset. The training set is used to train and optimize models. The testing set (unseen set) is used to evaluate models.

### 3.5. Machine Learning Approach

Five models are utilized in the ML approach: NB, KNN, DT, RF, and LR. As a feature extraction method, varied-sized n-grams coupled with TF-IDF are utilized upon ML models, which are optimized using grid search with cross-validation.

#### 3.5.1. Feature Extraction Method

The Bag-of-Words representation involves counting the number of times each word appears in a text. Every word has a number that denotes a column. Bag-of-word implementation done using TF-IDF with different n-gram sizes.

The context of the acquired words is preserved using the N-gram approach. It employs a collection of sequentially ordered words based on the value of the N variable. It could be uni-gram, bi-gram, and trig-gram if N = 1, N = 2, and if N = 3, respectively.
Uni-grams if the number of one word contained within a tweet.Bi-grams when the number of two-word sequences contained within a tweet.Tri-grams are the number of three-word sequences contained within a tweet.Four-gram is the number of four-word sequences contained within a tweet.
TF-IDF represents a statistical measure which applied to weight the importance “f” for each word with reference to the corpus. TF-IDF implementation includes two steps. Firstly, calculate the number of occurrences (TF) for each word presented in the document or tweet and then, Followed by finding each word occurrence frequency (IDF) throughout the whole document or tweet. Small values for the TF-IDF mean less significance for the word and vice versa. The larger values for TF-IDF mean the less frequent word in the corpus fixing its significance [44,45].
(1)$$\;\mathrm{Equation}\;\mathrm{to}\;\mathrm{measure}\;\mathrm{TF}\;=\frac{\;\mathrm{number}\;\mathrm{of}\;\mathrm{the}\;\mathrm{word}’s\;\mathrm{occurrence}\;}{\;\mathrm{total}\;\mathrm{number}\;\mathrm{of}\;\mathrm{the}\;\mathrm{words}\;\in \;\mathrm{the}\;\mathrm{document}\;}$$
(2)$$\mathrm{Equation}\;\mathrm{to}\;\mathrm{measure}\;\mathrm{IDF}\;=\log\frac{\;\mathrm{number}\;\mathrm{of}\;\mathrm{documents}\;}{\;\mathrm{number}\;\mathrm{of}\;\mathrm{documents}\;\mathrm{that}\;\mathrm{contain}\;\mathrm{the}\;\mathrm{word}\;}$$The TF-IDF is calculated by multiplying the TF with the IDF values for each word. If all of the words have equal weight, TF attempts to count their occurrences in the data. IDF, on the other hand, assigns weights to words based on their relevance and distinctiveness.

#### 3.5.2. Optimization Methods

A tuning procedure that aims to determine the ideal hyperparameter values. In Complex models that have diverse hyperparameters, It became necessary to figure out the optimal combination for the values of the hyperparameters through searching in a multi-dimensional space [46] In order to optimize ML models, grid search with stratified Cross-validation is utilized.

Grid search is a hyperparameter tuning approach based on partitioning the hyperparameter domain into discrete grids. Then, using cross-validation to determine the grid point that maximizes the average value in cross-validation [47]. This point indicates the best possible combination of hyperparameter values. Grid search determines the optimal point in the domain by traversing all possible combinations.Cross-validation is a statistical procedure for evaluating and comparing learning algorithms that involve dividing data into two segments: one for training a model and the other for validating the model [48] or randomly dividing the set of observations into roughly equal-sized k folds, or groups. The procedure’s general steps can be described as follows: shuffle the dataset at random first, then divide the dataset into k groups, with one group as a test data set and the remaining groups serving as training data sets. The model is then trained for the training set and tested on the test set. Finally, based on the sample of model evaluation scores, summarise the skill obtained by the model [49].

#### 3.5.3. ML Algorithms

The ML algorithms are briefly explained in this section.

Logistic Regression (LR) corresponds to supervised classification as a reliable and well-defined procedure [50]. It can be thought of as an extension of normal regression as it can only formulate variables that reflect the occurrence or non-occurrence of an event in general. The LR model assists in determining the likelihood of allocating a new instance to a specific class. Since the logistic regression model outputs predicted probability values that are mapped to two (binary classification) or more (multi-class classification) classes, a threshold should always be set to discriminate between them [51]. It permits us to model a correlation between a binary/binomial target variable and several predictor factors. The term “logistic” originates from the cost function (logistic function) with a form of Sigmund function with a distinctive S-shaped curve. Figure 2 shows an illustration of the LR boundary curve with its elements.The LR is a sigmoid function-based transformation of a linear regression. The likelihood for a specific categorization is represented on the vertical axis, while the value of *x* is represented on the horizontal axis. It is presumed that y∣x has a Bernoulli distribution. The formula of LR is as follows [50]:
(3)$$F\left(x\right)=\frac{1}{1+e^{-\left(\beta_{0}+\beta_{1}x\right)}}$$Here, $\beta_{0}+\beta_{l}x$ is comparable to the linear model y=ax+b The logistic function uses a Sigmund function to tie they value from a broad scale to a range of 0, 1. Multinomial logistic regression is a more generalized version of logistic regression that models a categorical variable with more than two values.Decision Tree (DT) is advantageous for organizing the data into a tree-like structure. The DT algorithm is one of the first machine learning algorithms. It is highly effective at classifying and filtering solutions in order to reach the best choices by comparing results for assorting data items into a tree-like structure [51,52].Starting with the root node, DT is usually built of multiple tiers. All internal nodes have at least one child and indicate input attribute or variable testing. The branching process repeats itself, directing the appropriate child node until it reaches the leaf node, depending on the results of the test that represents a decision [53]. An illustration of a DT with its elements and rules is depicted in Figure 3.Figure 3 shows each variable (A, B, and C) as a circle, while the decision outcomes (Class X and Class Y) are depicted as rectangles. Each branch is labeled with ‘True’ or ‘False’ depending on the outcome value from the test of its ancestor node on the route to successfully classify a sample to a class.Random Forest (RF) is a tree-based machine learning technique that combines the power of several decision trees to make the best possible conclusions [54]. The DT method is simple to interpret and comprehend. However, a single tree is insufficient to produce effective outcomes. This is why the RF algorithm has become so crucial [55].The RF is a classifier made up of various DTs. Overfitting can occur in some deep DTs, resulting in a lot of fluctuation in the classification results, even for modest changes in the input data [17]. The input vector of RF must be passed down to every DT in the forest. Then, each DT evaluates a separate component for that input vector, yielding a classification result. In the case of numeric classification, the forest picks the classification with the highest average for all trees in the forest or the most votes in the case of discrete classification. Each node in the decision tree operates on a subset of features to determine the output. To generate the final result, RF aggregates the output of the different decision trees. Figure 4 depicts a representation of the RF algorithm.The Naive Bayesian (NB) classifier is based on Bayes’ theorem, which maintains predictor independence assumptions [56,57]. Its goal is to describe the probability of an event based on past knowledge of the circumstances. Despite the possibility of dependency among the class members, it is assumed that a specific feature in a class is unrelated to any other feature. The Bayes theorem allows you to calculate posterior probability P(c∣x) from P(x), P(c) and P(c∣x). According to the NB classifier, the influence of predictor (*x*) on a given class (*c*) is unrestricted by the values of other predictors. Class conditional independence is the name given to this presumption.
(4)$$P\left(c|x\right)=\frac{P(x|c)P(c)}{P(c)}$$
where P(c∣x) represents the posterior probability of class (target) given predictor (attribute), P(x∣x) represents the eventuality, which is the probability of predictor given class, P(c) represents the prior probability of a class, which is equal to the occurrence of certain cases of *y* divided by the total number of records, and P(x) represents the prior probability of predictor.K-nearest neighbor (KNN) is one of the primary and straightforward ML algorithms, relying on the Supervised Learning approach [58]. The KNN algorithm assumes that the new attached case/data and the existing cases are comparable and then places the latest data in the appropriate category that is similar to the existing categories [51,59]. Figure 5 displays how the KNN works.

### 3.6. Deep Learning Approach

In the DL approach, three models: RNN, LSTM, and GRU. CBOW word embedding is used as the feature extraction method. The Keras-tuner is used to optimize the DL models.

#### 3.6.1. Optimization Method

KerasTuner is a salable hyperparameter optimization framework with built-in Hyperband, Bayesian Optimization, and Random Search algorithms to configure the search space with a define-by-run syntax, then leverage one of the available search algorithms to fetch the optimal hyperparameter values suitable for the models [60]. A set of hyperparameter values is adapted for RNN, LSTM, and GRU models. Each layer has a different amount of neurons, ranging from 100 to 1000. reg_rate for L2 regularizer is adapted to the values 0.0001, 0.0002, 0.0003, 0.0004, and 0.0005. The dropout value for the dropout layer is adapted between 0.1 to 0.9.

#### 3.6.2. Feature Extraction Method

Arabic contains a pre-trained distributed word embedding for the Arabic language that includes different word embedding models in Tweets and Wikipedia. In this paper, we used Twitter-CBOW with a 300 vector size.

#### 3.6.3. DL Algorithms

Figure 6 presents main layers of RNN, LSTM, and GRU models that consist of embedding, hidden, dropout, flatten, and output layers.

The embedding layer takes word embedding a matrix as input. It takes three arguments: input_dim represents the size of a word in tweets. output_dim defines the size of the output vectors from this layer for each word. It is adapted 300 because TwitterCBOW word embedding = 300 vector size. input_length represents the length of the input sequence = 20,000. RNN, LSTM, and GRU are used in the hidden layer. The dropout layer is a regularization technique that is used to reduce the overfitting and complexity of models. The output layer includes two neurons to predict the output of opinion, positive or negative. We used the softmax as an activation function; the ADAM optimizer has the value of learning_rate = 0.0004.

The DL algorithms are briefly explained in this section.

Recurrent Neural Network (RNN): When it comes to sequence data inputs, RNN is a type of neural network that works best with feedforward networks. We will need to modulate the neural network to recognise dependencies if we have sequence data where one data point is dependent on the previous data point. RNNs have the concept of “memory”, which enables them to store prior input states or data in constructing the sequence’s next output [61].RNN has a feedback cycle as shown in Figure 7 [62]. In order to get the shape illustrated in Figure 8, the feedback loop can be unrolled in three-time steps. The notation is as follows: At time step *t*, $x_{t}$ is a scalar number with a single characteristic. The network’s output at time step *t* is $y_{t}$. The values of the hidden units/states at time *t* are stored in the vector $y_{t}$. This is also known as the current context, and the $h_{0}$ vector is set to zero. In the recurrent layer, there are $w_{x}$ weights connected with inputs. Weights linked with hidden units in the recurrent layer are $w_{h}$. Weights related to concealed output units are $w_{y}$. The bias associated with the feedforward layer is $b_{h}$, which is the bias associated with the recurrent layer. We can unfold the network for *k* time steps in each time step to retrieve the output at time step *k* + 1. The unfolded network resembles the feedforward neural network in appearance. The operation depicted by the rectangle contained within the unfurled network. The activation function *f* [62]:
(5)$$h_{t+l}=f\left(x_{t},h_{t},w_{x},w_{k},b_{k}\right)=f\left(w_{x}x_{t}+w_{k}h_{t}+b_{b}\right)$$
(6)$$y_{t}=f\left(h_{t},w_{k}\right)=f\left(w_{k}\cdot h_{t}+b_{y}\right)$$Long Short Term Memory (LSTM): When dealing with short-term dependencies, RNNs perform admirably. When dealing with a large amount of irrelevant input, the RNN may need to retain the context, yet the relevant information may be disconnected from the moment where it is needed, causing the RNN would fail. In order to prevent the long-term dependence of RNNs, researchers proposed long short term memory neural networks. To do this, the core of LSTM is the cell state, which may add or delete information from cells while selectively allowing information to flow via the door mechanism. The LSTM is made up of three gates: forget gate, input gate, and output gate. As illustrated in Figure 9, Both the forget gate and the input gate determine which information is erased from and added to the cell state. When these two points are known, the cell state can be updated. Finally, the network’s final output is determined by the output gate [63].The state of each node in this process is determined by below equations [63].
(7)$$f_{t}=\sigma \left(W_{f}\cdot \left[h_{t-1},x_{t}\right]+b_{f}\right)$$
(8)$$i_{t}=\sigma \left(W_{i}\cdot \left[h_{t-1},x_{t}\right]+b_{i}\right)$$
(9)$${\overset{ˇ}{c}}_{t}=\tanh\left(W_{c}\cdot \left[h_{t-1},x_{t}\right]\right)+b_{c}$$
(10)$$c_{t}=f_{t}*c_{t-1}+i_{t}*{\overset{ˇ}{c}}_{t}$$
(11)$$o_{t}=\sigma \left(W_{0}\cdot \left[h_{t-1},x_{t}\right]+b_{0}\right)$$
(12)$$h_{t}=o_{t}*\tanh\left(c_{t}\right)$$$h_{t-1}$ denotes the previous layer’s hidden state, $x_{t}$ denotes the current input, *W* and *b* denote the weight and bias, Sigmund function, $f_{t}$ denotes the forget gate’s output, ${\overset{ˇ}{c}}_{t}$ denotes the input gate’s output, ${\overset{ˇ}{c}}_{t}$ denotes the temporary intermediate state, $c_{t-1}$ denotes the previous layer’s cell state, $c_{t}$ denotes the next layer’s cell state, $o_{t}$ denotes the output of the output gate, and $h_{t}$ denotes the hidden state of the next layer.Gated Recurrent UnitOne of the most common recurrent neural network types has seen a lot of use in machine translation. The Gated Recurrent Unit, or GRU, follows the same workflow as the RNN, with the exception of the operations and gates associated with each GRU unit. To capture dependencies on diverse time scales, GRU was used to produce each recurrent unit. The GRU, like the LSTM unit, has gating units that influence the flow of information inside the unit without the use of distinct memory cells, as shown in Figure 10. GRU integrates two gate operating techniques named Update gate and Reset gate to solve the problem presented by ordinary RNN [64].The GRU equations are defined below [65].
(13)$$z_{t}=\sigma_{g}\left(W_{z}x_{t}+U_{z}h_{t-1}+b_{z})\right.$$
(14)$$r_{t}=\sigma_{g}\left(W_{r}x_{t}+U_{r}h_{t-1}+b_{r})\right.$$
(15)$${\hat{h}}_{t}=\emptyset_{h}\left(W_{h}x_{t}+U_{h}\left(r_{t}⊙h_{t-1}\right)+b_{h}\right)$$
(16)$$h_{t}=\left(1-z_{t}\right)⊙h_{t-1}+z_{t}⊙{\hat{h}}_{t}$$
where *W*, *U*, and *b* are parameter matrices and vector [65] and $h_{t}$ is the output vector, ${\hat{h}}_{t}$ is the candidate activation vector, $z_{t}$ is the update gate vector, and $r_{t}$ is the reset gate vector. The update gate is in charge of ensuring that the precious memory is preserved in order to pass on to the next state. This is extremely useful since the model can choose to duplicate all of the data from the past, eliminating the risk of vanishing the gradient. The reset gateregulates how new input is incorporated into the previous memory. The reset gate is activated first, and it stores pertinent information from the previous time step into new memory content. The input vector and hidden state are then multiplied by their weights. The element-wise multiplication between the reset gate and the previously hidden state multiple is then calculated. After combining the preceding steps, the non-linear activation function is used to construct the next sequence.

### 3.7. The Proposed Stacking Ensemble Model

Figure 6 illustrates the proposed model’s architecture. Reveals that prediction performance can be improved by using an ensemble of multiple independently trained models. Different pre-trained models are loaded: RNN, GRU, and LSTM. Each model has its own architecture, and we freeze each layer of the model except the final layer (output layer). Then, each pre-trained model utilized the training set’s word embedding matrix for training models and generating predictions. The forecast of each model’s training set is then blended into the stacking. In stacking, we combine the training set predictions from each model and send them to a meta-learner to learn. In addition, we integrate each model’s prediction of the testing set in stacking and end it in the meta-learner to make the final prediction.

### 3.8. Evaluating Models

The suggested models are evaluated using four common performance metrics: accuracy (ACC), precision (PRC), recall (REC), and F1-score (F1). These are calculated in the following [66]:(17)$$\mathrm{Accuracy}=\frac{\mathrm{TP}\;+\;\mathrm{TN}}{\mathrm{TP}\;+\;\mathrm{FP}\;+\;\mathrm{TN}\;+\;\mathrm{FN}}.$$
(18)$$\mathrm{Precision}=\frac{\mathrm{TP}}{\mathrm{TP}\;+\;\mathrm{FP}}$$
(19)$$\mathrm{Recall}=\frac{\mathrm{TP}}{\mathrm{TP}\;+\;\mathrm{FN}}$$
(20)$$F1=\frac{2\cdot \mathrm{precision}\;\cdot \mathrm{recall}}{\mathrm{precision}+\mathrm{recall}}$$

## 4. Experiments Results

This section presents the experimental setup, and results of each datasets.

### 4.1. Experimental Setup

The experiments for this study were carried out using Python on a Google Colab. The sci-kit-learn and Keras packages were used to implement ML and DL models, respectively. Grid-search with stratified cross-validation and the KerasTuner package was also used to optimize ML and DL models, respectively. The proposed model was implemented using the sci-kit-learn and Keras packages. The meta-learners were tuned via grid search. Three Arabic sentiment analysis benchmark datasets were separated into training and testing sets with 80% training and 20% testing. The cross-validation and testing performance results will be recorded. Learning rate of admin optimizer: 0.0001, Batch size: 1500, and Epochs: 20 were some of the values of DL model parameters that were altered. The KerasTuner method library has chosen the best settings for the parameters of the DL models for each dataset, as shown in Table 1.

### 4.2. Results of ASTC Dataset

This part is presented the performance results of ML and DL and The proposed model with cross-validation and testing results over the ASTC dataset. And it is presented the best values of DL parameters models that have been selected by the KerasTuner method.

Table 2 shows the values of four metrics, including ACC, PRC, REC, and F1 of cross-validation and testing results for ML models, DL models, and the proposed model.

#### 4.2.1. Cross-Validation Results

When comparing the performance results of ML techniques, it is clear that LR with uni-gram is the best classifier (ACC = 93.21 percent, PRE = 93.23%, REC = 93.21%, and F1 = 93.2%) when compared to other ML classifiers. On the other hand, DT with four-gram had the worst performance compared to other classifiers (ACC = 60.97%, PRE = 72.27%, REC = 60.95%, and F1 = 54.29%). In terms of (ACC = 92.79%, PRE = 92.87%, REC = 92.85%, and F1 = 92.78%), RF with uni-gram has the best second performance. In comparison, DT has the lowest performance in terms of (ACC = 87.51%, PRE = 87.63%, REC = 87.51%, and F1 = 87.5%) according to uni-gram. Also, because the number of times a single word is repeated is greater than the frequency of many words together, we can see that unigram with TF-IDF feature extraction achieves the best overall performance in the classifiers.

According to DL models, LSTM has the highest result in terms (ACC = 96.96%, PRE = 97.0%, REC = 97.0%, and F1 = 97.0%), and it improved ACC by 3.75%, PRE by 3.77%, and REC by 3.79% and F1 by 3.8% compared to LR with unigram. While RNN and GRU are registered the same approximately performance.

According to the proposed model, stacking LR and stacking SVM achieved the highest performance compared to ML models and DL models. The stacking LR is registered performance in terms (ACC = 98.08%, PRE = 98.09%, REC = 98.08%, and F1 = 98.08%), and it improved ACC by 1.12%, PRE by 1.09%, and REC by 1.8% and F1 by 1.08% compared to LSTM models.

#### 4.2.2. Testing Results

When comparing the performance results of ML approaches, it is apparent that LR with unigram is the best classifier compared with other ML classifiers in terms of (ACC = 91.04%, PRE = 91.12%, REC = 91.04%, and F1 = 91.03%). In contrast, DT with four-gram achieved the worst performance compared with different classifiers in terms of (ACC = 60.39%, PRE = 72.36%, REC = 60.39%, and F1 = 53.44%). RF with unigram achieves the best second performance with unigram in terms of (ACC = 90.68%, PRE = 90.72%, REC = 90.68%, and F1 = 90.67%). Also, we can be observed that unigram with TF-IDF feature selection is achieved the best performance overall in the classifiers because the number of times a single word is repeated will be more than the frequency of more than one word together.

According to DL models, LSTM is achieved the highest result in terms (ACC = 91.89%, PRE = 91.9%, REC = 91.89%, and F1 = 91.89%) and it improved ACC by 0.85%, PRE by 0.78%, and REC by 0.85% and F1 by 0.86% compared to LR with unigram. While GRU is obtained the lowest performance in terms (ACC = 88.6%, PRE = 88.61%, REC = 88.6%, and F1 = 88.6%).

According to the proposed model, stacking LR and stacking SVM achieved the highest performance compared to ML and DL models. The Stacking LR is registered performance in terms (ACC = 92.22%, PRE = 92.23%, REC = 92.22%, and F1 = 92.22%) and it improved ACC by 0.33%, PRE by 0.33%, and REC by 0.33% and F1 by 0.33% compared to LSTM models.

### 4.3. Results of ArTwitter Dataset

This part is presented the performance results of ML and DL and The proposed model with cross-validation and testing results over the ArTwitter dataset. Table 3 shows the values of four metrics, including ACC, PRC, REC, and F1 of cross-validation and testing results for ML models, DL models, and the proposed model.

#### 4.3.1. Cross-Validation Results

When comparing the performance results of ML approaches, it is apparent that LR with unigram is the best classifier compared with other ML classifiers in terms of (ACC = 85.38%, PRE = 85.68%, REC = 85.38%, and F1 = 85.37%). In contrast, KNN with tri-gram achieved the worst performance compared with different classifiers in terms of (ACC = 48.97%, PRE = 31.17%, REC = 48.97%, and F1 = 32.61%). NB with unigram achieves the best second performance in terms of (ACC = 84.23%, PRE = 85.1%, REC = 84.23%, and F1 = 84.1%). In comparison, based on unigram, DT performs the lowest performance in terms of (ACC = 76.79%, PRE = 77.49%, REC = 76.28%, and F1 = 77.73%). Also, we can be observed that unigram with TF-IDF feature selection is achieved the best performance overall in the classifiers because the number of times a single word is repeated will be more than the frequency of more than one word together.

According to DL models, GRU has the highest result (ACC = 88.48%, PRE = 88.48%, REC = 88.48%, and F1 = 88.48%). It improved ACC by 3.1%, PRE by 2.8%, REC by 3.1%, and F1 by 3.11% compared to LR with unigram. While LSTM and RNN are registered the same approximately performance.

According to the proposed models, stacking SVM is achieved the highest performance compared to ML and DL models in terms (ACC = 92.24%, PRE = 92.39%, REC = 92.24%, and F1 = 92.24%), and it improved ACC by 3.76%, PRE by 3.91%, and REC by 3.76% and F1 by 3.76% compared to GRU models.

#### 4.3.2. Testing Results

When comparing the performance results of ML approaches, it is apparent that LR with unigram is the best classifier compared with other ML classifiers in terms of (ACC = 75.7%, PRE = 75.99%, REC = 75.7%, and F1 75.67%). In contrast, KNN has the worst performance compared with different classifiers based on ACC, PRE, and REC. Based on the F1 term, the NB with Four-gram has performed the worst performance at 31%. NB with unigram achieves the best second performance with unigram in terms (ACC = 74.17%, PRE = 75.68%, REC = 74.17%, and F1 = 73.7%). Also, we can be observed that unigram with TF-IDF feature selection is achieved the best performance overall in the classifiers because the number of times a single word is repeated will be more than the frequency of more than one word together.

According to DL models, GRU has the highest result in terms (ACC = 82.1%, PRE = 82.63%, REC = 82.1%, and F1 = 82.05%) improved ACC by 6.4%, PRE by 6.64%, and REC by 6.4% and F1 by 6.38% compared to LR with unigram. While LSTM and RNN are registered the same approximately performance.

According to the proposed models, stacking SVM is achieved the highest performance compared to ML models and DL models in terms (ACC = 83.12%, PRE = 83.85%, REC = 83.12%, and F1 = 83.06%), and it improved ACC by 1.02%, PRE by 1.22%, and REC by 1.02% and F1 by 1.01% compared to GRU models.

### 4.4. Results of AJGT Dataset

This part is presented the performance results of ML and DL and the proposed model with cross-validation and testing results over the AJGT dataset. Table 4 shows the values of four metrics, including ACC, PRC, REC, and F1 of cross-validation and testing results for ML models, DL models, and the proposed model.

#### 4.4.1. Cross-Validation Results

When comparing the performance results of ML approaches, it is apparent that LR with unigram is the best classifier compared with other ML classifiers in terms of (ACC = 84.03%, PRE = 84.5%, REC = 84.03%, and F1 = 83.96%). In contrast, KNN with tri-gram is achieved the worst performance compared with other classifiers in terms of (ACC = 50.0%, PRE = 25.0%, REC = 50.0%, and F1 = 33.33%). NB with unigram achieves the best second performance with unigram in terms of (ACC = 83.47%, PRE = 83.85%, REC = 83.47%, and F1 = 83.43%). Also, we can be observed that unigram with TF-IDF feature selection is achieved the best performance overall in the classifiers because the number of times a single word is repeated will be more than the frequency of more than one word together.

According to DL models, LSTM has the highest result in terms (ACC = 93.4%, PRE = 93.5%, REC = 93.4%, and F1 = 93.4%), and it improved ACC by 9.55%, PRE by 9.08%, and REC by 9.55% and F1 by 9.62% compared to LR with unigram. While RNN is registered the lowest performance in terms (ACC = 86.86%, PRE = 86.86%, REC = 86.86%, and F1 = 86.86%).

According to the proposed ensemble model, LR stacking and SVM stacking achieved the highest performance compared to ML models and DL models. LR is registered performance in terms (ACC = 93.4%, PRE = 93.5%, REC = 93.4%, and F1 = 93.4%) and it improved ACC by 3.95%, PRE by 4.05%, and REC by 3.95% and F1 by 3.95% compared to LSTM models.

#### 4.4.2. The Testing Results

When comparing the performance results of ML approaches, it is apparent that LR with unigram is the best classifier compared with other ML classifiers in terms of (ACC = 76.94%, PRE = 77.25%, REC = 76.94%, and F1 = 76.88%). In contrast, KNN with four-gram achieved the worst performance compared with other classifiers in terms of (ACC = 50.28%, PRE = 75.07%, REC = 50.28%, and F1 = 33.95%). NB with unigram achieves the best second performance with unigram in terms of (ACC = 76.94%, PRE = 77.01%, REC = 76.94%, and F1 = 76.93%). Also, we can be observed that unigram with TF-IDF feature selection is achieved the best performance overall in the classifiers because the number of times a single word is repeated will be more than the frequency of more than one word together.

According to DL models, LSTM is achieved the highest result in terms ACC = 84.72%, PRE = 84.9%, REC = 84.72%, and F1 = 84.7%) and it improved ACC by 7.78%, PRE by 7.65%, and REC by 7.78% and F1 by 7.82% compared to LR with unigram. While RNN is registered the lowest performance in terms (ACC = 82.78%, PRE = 82.78%, REC = 82.78%, and F1 = 82.74%).

According to the proposed model, LR stacking and SVM stacking achieved the highest performance compared to ML models and DL models. LR is registered performance in terms (ACC = 86.11%, PRE = 86.11%, REC = 86.11%, and F1 = 86.11%) and it improved ACC by 3.33%, PRE by 3.09%, and REC by 3.33% and F1 by 3.37% compared to LSTM models.

## 5. Discussion

Overall, compared to other algorithms, the proposed model has the highest performance for cross-validation and testing results for all datasets.

For ASTC dataset, the comparison between the highest performing models for cross-validation and testing for ASTC dataset is shown Figure 11. The proposed model (stacking LR) has the highest performance in terms (ACC = 98.08%, PRE = 98.09%, REC = 98.08%, F1 = 98.08%) and (ACC = 92.22%, PRE = 92.23%, REC = 92.22%, F1 = 92.22%) for cross validation and testing; respectively. while LR with Unigram has the lowest performance in terms (ACC = 93.21%, PRE = 93.23%, REC = 93.23%, F1 = 93.2%) and (ACC = 91.04%, PRE = 91.12%, REC = 91.04%, F1 = 91.03%) for cross validation and testing, respectively.

For ArTwitter dataset, the comparison between the highest performing models for cross-validation and testing for ArTwitter is shown Figure 12. the proposed model (SVM stacking) has the highest performance in terms(ACC = 92.24%, PRE = 92.39%, REC = 92.24%, F1 = 92.24%) and (ACC = 83.12%, PRE = 83.85%, REC = 83.12%, F1 = 83.06%) for cross validation and testing; respectively. While SVM with unigram has the lowest performance in terms (ACC = 85.38%, PRE = 85.68%, REC = 85.38%, F1 = 85.37%) and (ACC = 75.7%, PRE = 75.99%, REC = 75.7%, F1 = 75.69%) for cross validation and testing, respectively.

For AJGT dataset, the comparison between the highest performing models for cross-validation and testing for AJGT dataset is shown Figure 13. The proposed model (stacking LR) has the highest performance in terms (ACC = 93.4%, PRE = 93.5%, REC = 93.4%, F1 = 93.4%) and (ACC = 86.11%, PRE = 86.13%, REC = 86.11%, F1 = 86.11%) for cross validation and testing; respectively. While LR with unigram has the lowest performance in terms (ACC = 84.03%, PRE = 84.5%, REC = 84.03%, F1 = 83.96%) and (ACC = 76.94, PRE = 77.25%, REC = 76.94%, F1 = 76.88%) for cross validation and testing, respectively.

The proposed models are compared with the literature studies. Table 5 compares the proposed model to existing literature for the three datasets. We noticed that the proposed models for Arabic sentiment analysis perform better than other existing works that use methods based on DL and ensemble learning. First, compared to the authors who used the ArTwitter dataset, in [26], ACC was registered 88%, in [27], ACC was registered 92.39%, in authors used a voting algorithm and achieved ACC of 86%. The authors [28] used CNN-LSTM hybrid model and the performance was 86.45%, 86.46%, 86.45%, and 86.45% for ACC, PRE, REC, and F1, respectively. For the AJGT dataset, the authors in [67] proposed an SVM-based model and achieved 88.72% for ACC and 88.27% for F1.

In summary, the proposed models have registered the highest cross-validation and testing performance with the three datasets. Moreover, we compared our models with the literature studies that have used other datasets than the used three datasets. For instance, authors in [36] explored the ensemble models of voting, bagging, boosting, stacking, and RF. They achieved an F1 of 85% using the Arabic Tweets dataset. As clearly noticed, our models achieved better results compared to [36].

In addition, many studies shared the code of their implemented models. In [68], the authors used DL techniques, including RNN, LSTM, Bi-LSTM, and CNN, and two-word embedding techniques Word2Vec and fastText for predicting the Sinhala Language Sentiment Analysis (https://github.com/LahiruSen/sinhala-sentiment-anlaysis-tallip (accessed on 2 April 2022)). In [69], the authors proposed a hybrid model based on CNN and LSTM and made comparisons against CNN, LSTM, and RNN for predicting the English Language sentiment analysis (https://github.com/pmsosa/CS291K (accessed on 2 April 2022)). The results show that their hybrid model was achieved the best performance. In [70], the authors evaluated CNN models using a different word embedding Word2Vec and Glove on different English sentiment analysis datasets (https://github.com/shreydesai/cnn-sentiment-analysis (accessed on 2 April 2022)).

## 6. Conclusions

This study proposes an optimized stacked ensemble DL model for solving the Arabic sentiment analysis problem. The proposed model combines three heterogeneous pre-trained DL models including RNN, LSTM, and GRU. We explore three meta-learners including LR, RF, and SVM. The meta-learner models are optimized using the grid search hyperparameter optimization technique. We tested the models on three well-known Arabic sentiment analysis benchmark datasets including AJGT, ASTC, and ArTwitter. The data splitting strategy is 80% training set and 20% testing set. The study compares the proposed models’ results with other regular ML and DL models. The utilized ML models include NB, KNN, DT, RF, and LR. The TF-IDF with variable-sized n-grams is used as a feature selection method for the ML models. The feature extraction approach is CBOW word embedding. The Keras-tuner is used to optimize the DL models. For ML models, the best results are achieved with the unigram, while the worst results are with the four-gram in all comparisons. The proposed model achieves the best results against all other ML and DL models in all comparisons including testing and cross-validation stages.

## Figures and Tables

**Figure 1 sensors-22-03707-f001:**
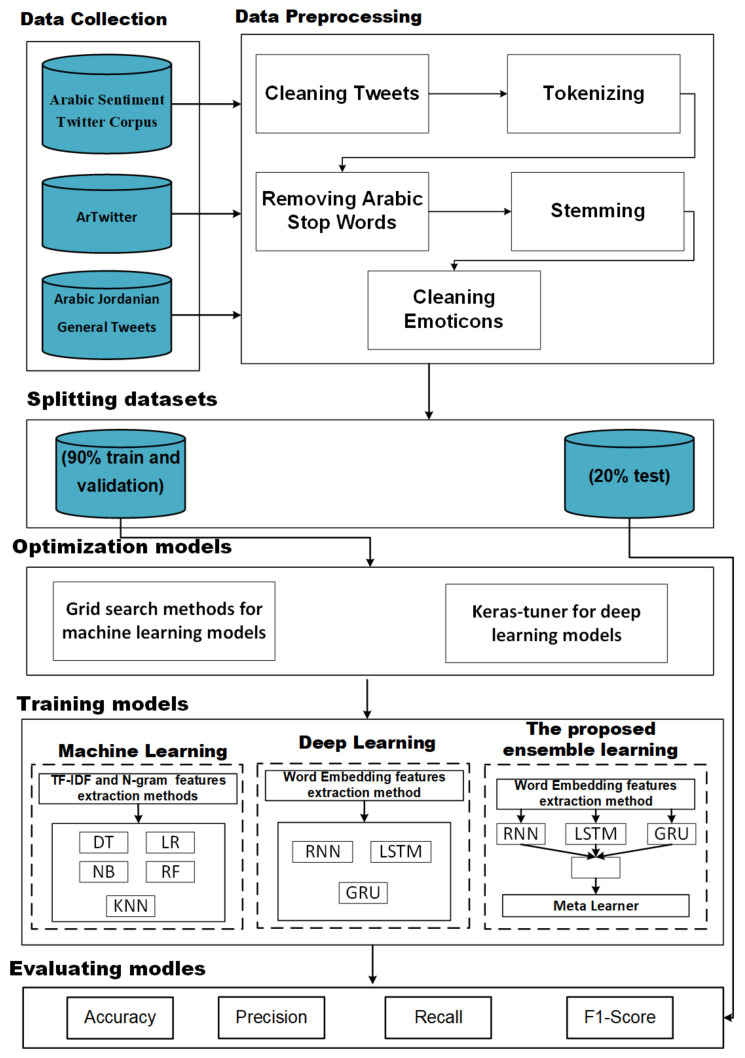
Steps of the predicting sentiment analysis for Arabic data.

**Figure 2 sensors-22-03707-f002:**
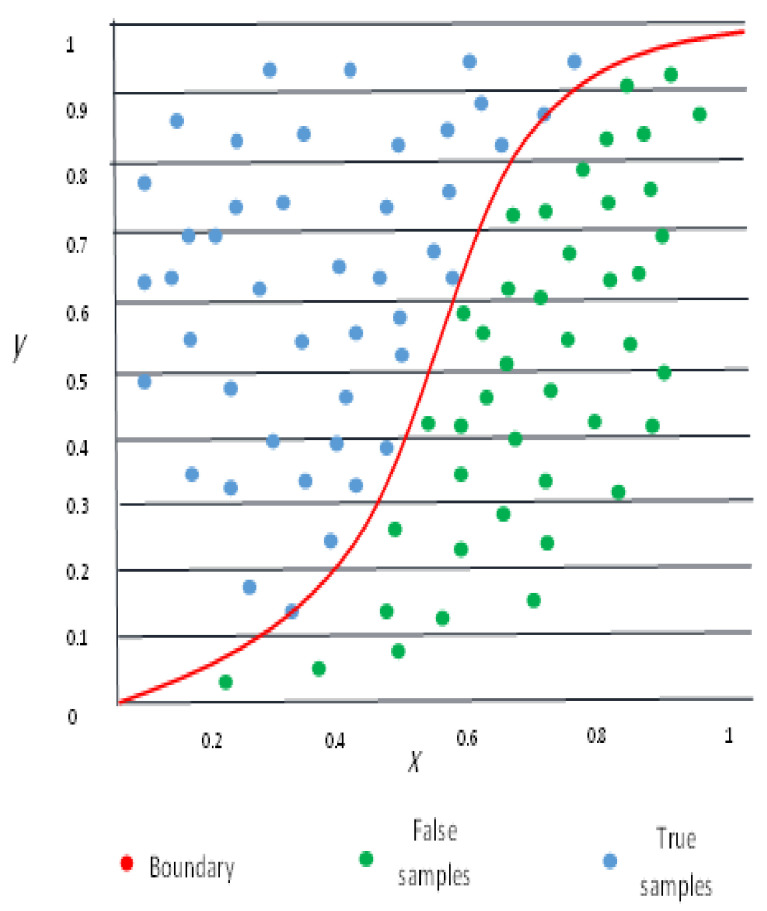
The logistic regression boundary curve.

**Figure 3 sensors-22-03707-f003:**
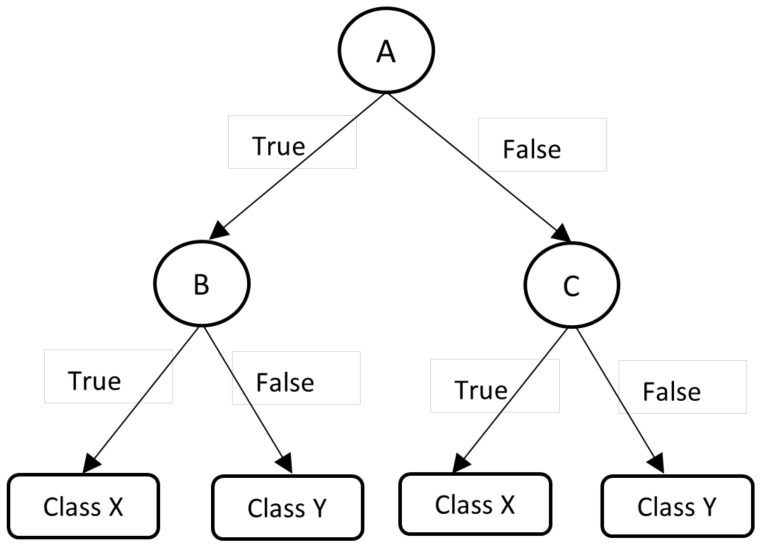
An illustration of the Decision tree.

**Figure 4 sensors-22-03707-f004:**
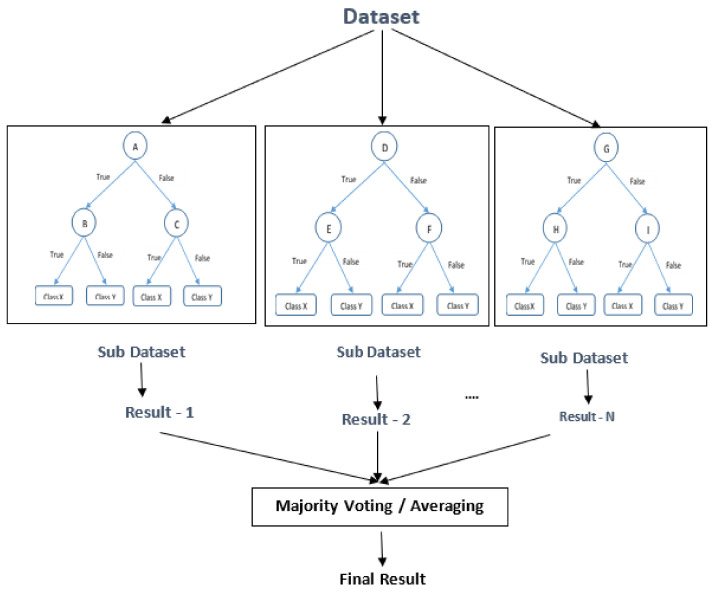
The RF which consists of three different decision trees. Each one was trained using a subset of the training dataset.

**Figure 5 sensors-22-03707-f005:**
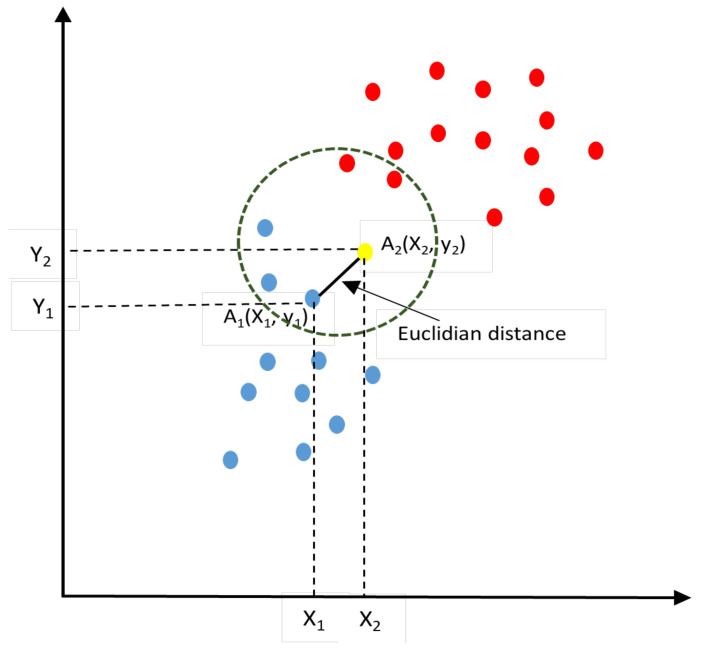
The K-nearest neighbor diagram.

**Figure 6 sensors-22-03707-f006:**
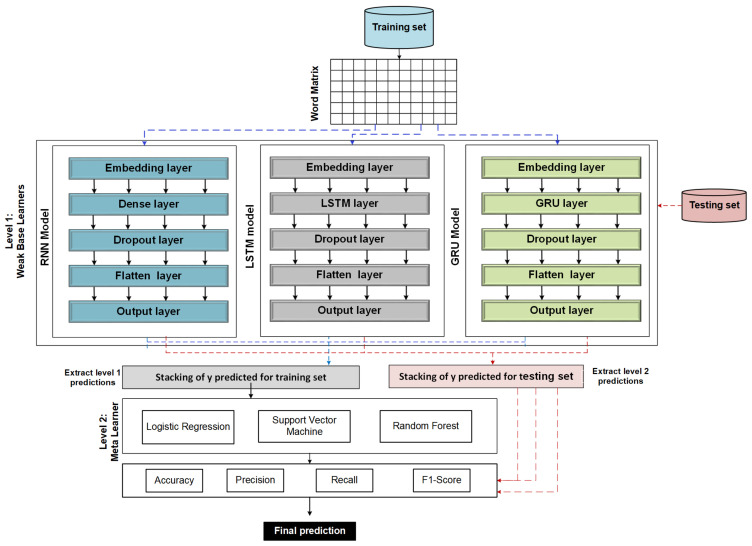
The architecture of the proposed stacking ensemble model.

**Figure 7 sensors-22-03707-f007:**
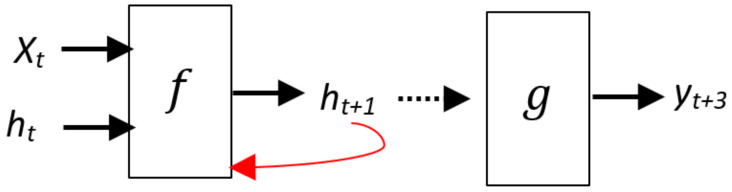
Compressed representation for the RNN [62].

**Figure 8 sensors-22-03707-f008:**
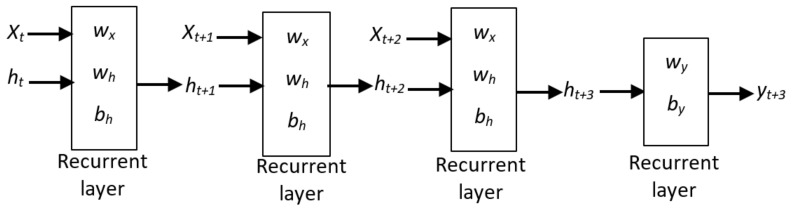
Unfolded network representation for the the RNN [62].

**Figure 9 sensors-22-03707-f009:**
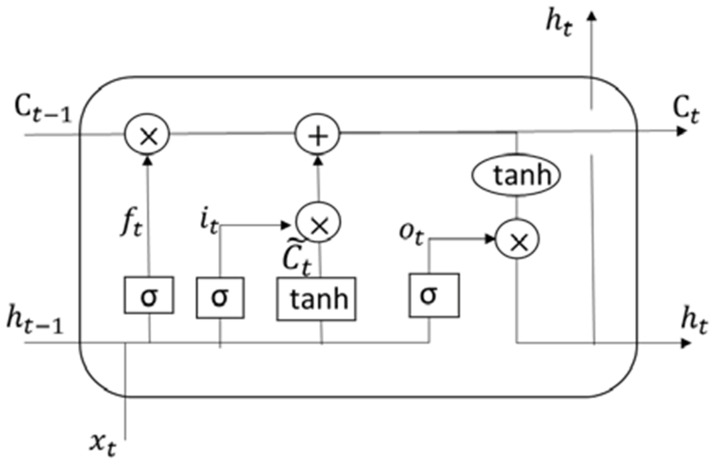
Representation for the Long Short Term Memory.

**Figure 10 sensors-22-03707-f010:**
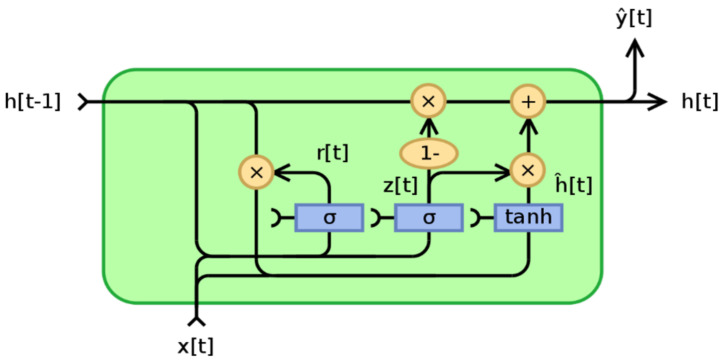
Illustrate Gated Recurrent Unit.

**Figure 11 sensors-22-03707-f011:**
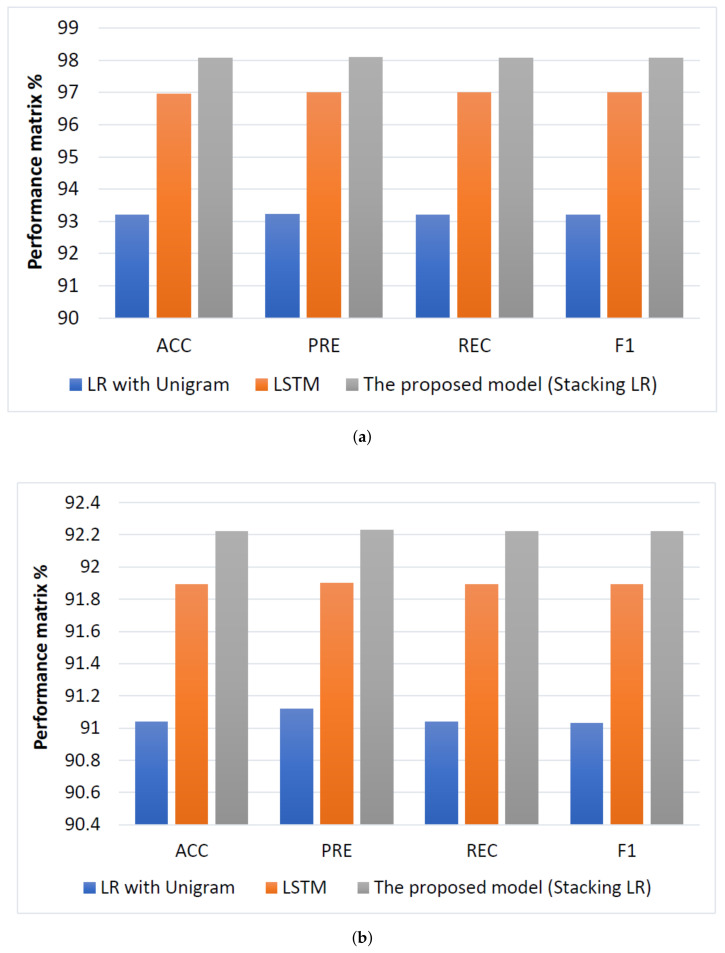
Comparison of performance the best models for ASTC dataset, (**a**) Cross-validation performance and (**b**) testing performance.

**Figure 12 sensors-22-03707-f012:**
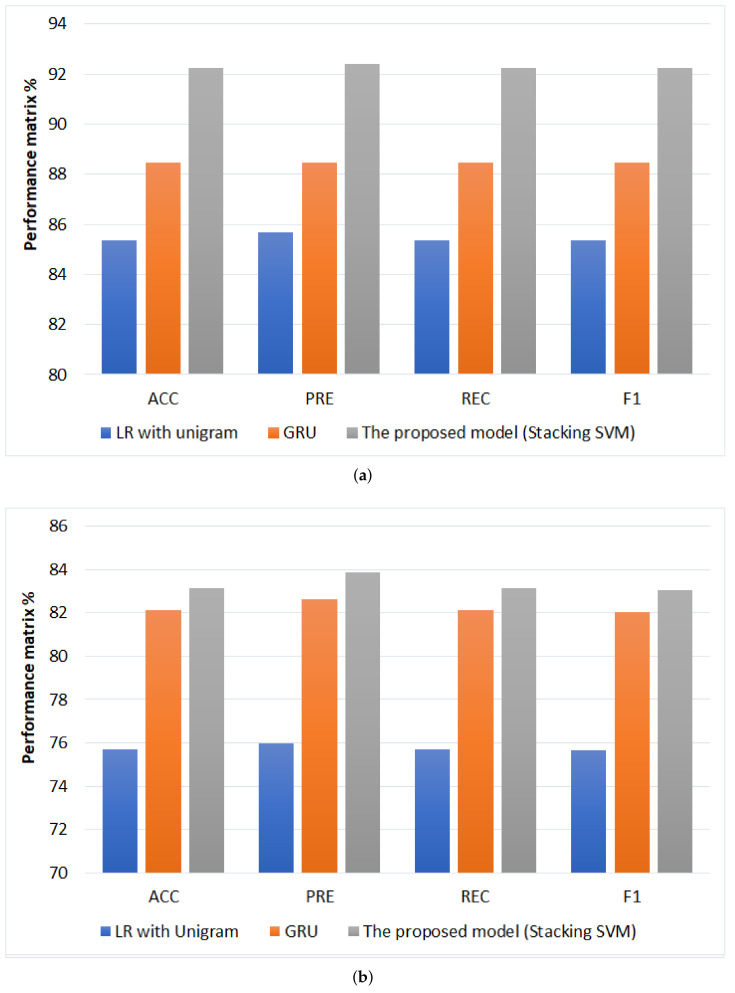
Comparison of performance the best models for ArTwitter, (**a**) Cross-validation performance and (**b**) testing performance.

**Figure 13 sensors-22-03707-f013:**
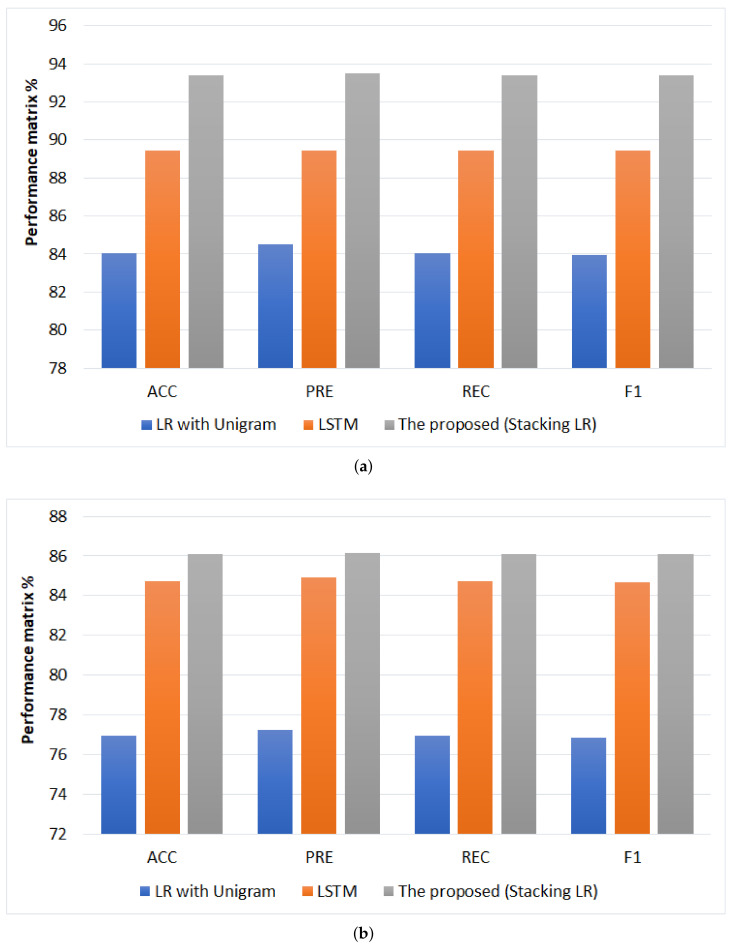
Comparison of performance the best models for AJGT dataset, (**a**) Cross-validation performance and (**b**) testing performance.

**Table 1 sensors-22-03707-t001:** The best values parameters of DL models for each dataset.

Dataset	Models	Neurons	Dropout	reg_rate1
ASTC dataset	RNN	300	0.7	0.0001
	LSTM	150	0.6	0.01
	GRU	300	0.2	0.4
ArTwitter dataset	RNN	1000	0.4	0.7
	LSTM	950	0.2	0.2
	GRU	500	0.4	0.05
AJGT dataset	RNN	500	0.5	0.4
	LSTM	400	0.8	0.0006
	GRU	750	0.8	0.05

**Table 2 sensors-22-03707-t002:** The performance results of ML, DL, and the proposed models for ASTC dataset.

Approach Models	Models	Matrix Size	Cross Validation Performance				Test Performance			
			ACC	PRE	REC	F1	ACC	PRE	REC	F1
ML models	DT	Unigram	92.16	92.13	92.14	92.12	89.55	89.57	89.55	89.55
		Bi-gram	77.54	82.83	77.56	76.61	75.97	81.11	75.97	74.96
		Tri-gram	73.67	77.13	63.68	58.55	62.93	63.15	62.93	62.75
		Four-gram	60.97	72.27	60.95	54.29	60.39	72.36	60.39	53.44
	KNN	Unigram	90.72	90.72	90.72	90.72	88.58	88.6	88.58	88.57
		Bi-gram	83.45	83.78	83.45	83.41	69.79	74.54	69.79	68.31
		Tri-gram	72.24	81.02	72.24	70.17	66.92	67.11	66.92	66.85
		Four-gram	71.27	74.89	71.27	70.33	64.3	64.63	64.3	64.12
	LR	**Unigram**	**93.21**	**93.23**	**93.21**	**93.2**	**91.04**	**91.12**	**91.04**	**91.03**
		Bi-gram	86.82	87.42	86.82	86.77	80.79	82.54	80.79	80.54
		Tri-gram	79.1	83.38	79.1	78.43	70.81	78.64	70.81	68.71
		Four-gram	75.23	82.05	75.23	73.87	67.08	78.13	67.08	63.57
	RF	Unigram	92.79	92.87	92.85	92.78	90.68	90.72	90.68	90.67
		Bi-gram	78.28	82.86	78.53	78.26	78.18	78.19	78.18	78.18
		Tri-gram	66.02	76.97	66.03	62.25	66.78	69.37	66.78	65.59
		Four-gram	63.35	75.94	63.41	58.26	63.83	69.36	63.83	60.96
	NB	Unigram	87.51	87.63	87.51	87.5	86.09	86.13	86.09	86.09
		Bi-gram	86.15	86.67	86.15	86.1	78.97	79.98	78.97	78.78
		Tri-gram	78.95	82.7	78.95	78.34	67.91	68.95	67.91	67.43
		Four-gram	74.76	78.02	74.76	74.03	64.05	66.79	64.05	62.45
DL models	RNN	CBOW	94.92	94.92	94.92	94.92	90.18	90.18	90.18	90.17
	**LSTM**	**CBOW**	**96.96**	**97.0**	**97.0**	**97.0**	**91.89**	**91.9**	**91.89**	**91.89**
	GRU	CBOW	94.89	94.87	94.87	94.87	88.6	88.61	88.6	88.6
The proposed model	**Stacking** **LR**	**CBOW**	**98.08**	**98.09**	**98.08**	**98.08**	**92.22**	**92.23**	**92.22**	**92.22**
	Stacking SVM	CBOW	98.07	98.07	98.07	98.07	92.1	92.11	92.1	92.1
	Stacking RF	CBOW	97.27	97.28	97.27	97.27	91.98	91.99	91.98	91.98

**Table 3 sensors-22-03707-t003:** The performance results of ML, DL, and the proposed model for dataset ArTwitter.

Approach Models	Models	Matrix Size	Cross-Validation Performance				Test Performance			
			ACC	PRE	REC	F1	ACC	PRE	REC	F1
ML models	DT	Unigram	76.79	77.49	76.28	77.73	72.63	73.67	72.63	72.41
		Bi-gram	60.13	76.05	60.32	53.15	56.27	75.18	56.27	46.81
		Tri-gram	53.08	70.64	53.33	38.82	52.17	75.34	52.17	37.12
		Four-gram	52.69	70.2	52.63	38.16	51.92	75.28	51.92	36.57
	KNN	Unigram	79.42	79.84	79.42	79.36	66.5	69.88	66.5	64.77
		Bi-gram	51.22	73.0	51.22	36.92	49.62	75.13	49.62	33.47
		Tri-gram	48.97	31.17	48.97	32.61	51.41	75.14	51.41	35.46
		Four-gram	49.36	35.3	49.36	35.96	50.64	50.18	50.64	45.81
	LR	**Unigram**	**85.38**	**85.68**	**85.38**	**85.37**	**75.7**	**75.99**	**75.7**	**75.67**
		Bi-gram	68.27	77.41	68.27	65.6	56.78	74.04	56.78	47.97
		Tri-gram	53.97	74.64	53.97	40.95	52.17	75.34	52.17	37.12
		Four-gram	53.4	75.68	53.4	39.61	52.17	75.34	52.17	37.12
	RF	Unigram	78.78	80.44	78.91	78.6	73.15	74.0	73.15	72.97
		Bi-gram	60.71	75.62	60.51	54.84	57.03	75.55	57.03	48.15
		Tri-gram	53.33	75.66	53.46	39.58	52.17	75.34	52.17	37.12
		Four-gram	53.33	75.63	53.33	39.23	52.17	75.34	52.17	37.12
	NB	Unigram	84.23	85.1	84.23	84.1	74.17	75.68	74.17	73.7
		Bi-gram	60.83	75.18	60.83	53.78	55.5	66.49	55.5	45.7
		Tri-gram	53.97	74.64	53.97	40.95	52.17	75.34	52.17	37.12
		Four-gram	53.4	75.68	53.4	39.61	51.17	74.34	51.17	31.12
DL models	RNN	CBOW	87.12	87.12	87.12	87.12	81.86	81.88	81.86	81.77
	LSTM	CBOW	87.83	87.83	87.83	87.83	81.33	81.6	81.33	81.27
	**GRU**	**CBOW**	**88.48**	**88.48**	**88.48**	**88.48**	**82.1**	**82.63**	**82.1**	**82.05**
The proposed model	Stacking LR	CBOW	91.99	92.07	91.99	91.99	82.35	82.93	82.35	82.3
	**Stacking SVM**	**CBOW**	**92.24**	**92.39**	**92.24**	**92.24**	**83.12**	**83.85**	**83.12**	**83.06**
	Stacking RF	CBOW	92.12	92.2	92.12	92.11	82.86	83.14	82.86	82.85

**Table 4 sensors-22-03707-t004:** The performance results of ML, DL, and the proposed models for AJGT dataset.

Approach Models	Models	Matrix Size	Cross Validation Performance				Test Performance			
			ACC	PRE	REC	F1	ACC	PRE	REC	F1
ML models	DT	Unigram	78.82	79.1	77.92	79.28	71.39	72.44	71.39	71.05
		Bi-gram	60.76	75.31	60.97	53.26	56.39	69.18	56.39	47.66
		Tri-gram	51.53	70.41	51.39	36.45	51.11	75.28	51.11	35.76
		Four-gram	50.0	35.05	50.07	33.64	50.0	25.0	50.0	33.33
	KNN	Unigram	78.26	79.3	78.26	78.06	68.89	70.09	68.89	68.42
		Bi-gram	51.18	60.19	51.18	39.84	50.28	51.16	50.28	38.6
		Tri-gram	50.0	25.0	50.0	33.33	50.28	75.07	50.28	33.95
		Four-gram	50.07	27.67	50.07	34.23	50.0	25.0	50.0	33.33
	LR	**Unigram**	**84.03**	**84.5**	**84.03**	**83.96**	**76.94**	**77.25**	**76.94**	**76.88**
		Bi-gram	67.99	76.69	67.99	65.11	56.94	67.97	56.94	49.14
		Tri-gram	53.54	75.21	53.54	40.79	51.11	75.28	51.11	35.76
		Four-gram	50.56	53.47	50.56	34.66	50.0	25.0	50.0	33.33
	RF	Unigram	78.26	80.81	78.96	78.07	75.83	77.56	75.83	75.45
		Bi-gram	60.76	76.34	60.56	53.59	56.94	68.83	56.94	48.88
		Tri-gram	52.43	65.32	52.29	38.32	51.11	75.28	51.11	35.76
		Four-gram	50.42	50.09	50.49	33.94	50.0	50.0	50.0	47.92
	NB	Unigram	83.47	83.85	83.47	83.43	76.94	77.01	76.94	76.93
		Bi-gram	60.9	72.04	60.9	54.67	56.94	67.97	56.94	49.14
		Tri-gram	53.54	75.21	53.54	40.79	51.11	75.28	51.11	35.76
		Four-gram	50.56	53.47	50.56	34.66	50.0	25.0	50.0	33.33
DL models	RNN	CBOW	86.86	86.86	86.86	86.86	82.78	83.04	82.78	82.74
	**LSTM**	**CBOW**	**89.45**	**89.45**	**89.45**	**89.45**	**84.72**	**84.9**	**84.72**	**84.7**
	GRU	CBOW	89.01	89.01	89.01	89.01	84.72	84.9	84.72	84.7
The proposed ensemble model	**Stacking** **LR**	**CBOW**	**93.4**	**93.5**	**93.4**	**93.4**	**86.11**	**86.13**	**86.11**	**86.11**
	Stacking SVM	CBOW	93.4	93.48	93.4	93.4	86.01	86.01	86.01	86.01
	Stacking RF	CBOW	92.9	93.05	92.99	92.98	85.83	85.89	85.83	85.83

**Table 5 sensors-22-03707-t005:** The comparison of results to previous studies.

Paper	Alg.	Dataset	Performance
Alayba et al. [26]	CNN+LSTM	ASTD	77% of ACC
		ArTwitter	88% of ACC
Hanane Elfaik [27]	Bi-LSTM	ASTD	76.83% of ACC
		ArTwitter	92.39% of ACC
Al-Saqqa et al. [36]	voting algorithm based on KNN, NB, DT, and SVM	ArTwitter	86% of ACC
Al-Azani et al. [28]	CNN and LSTM	ArTwitter	86.45% of ACC 86.46% of PRE 86.45% of REC 86.45% of F1
Alomari et al. [67]	SVM	AJGT	88.72% of ACC 88.27% of F1
Al-Azani et al. [37]	Voting, Bagging, Boosting, Stacking and RF	Arabic Tweets	85% of F1
The proposed stacking model	The pre-trained RNN, GRU and LSTM with meta-learner LR	ASTC dataset	For cross-validation, 98.08%of ACC, 98.09% of PRE, 98.08% of REC, 98.08% of F1
			For testing, 92.22% of ACC, 92.23% of PRE, 92.22% of REC, 92.22% of F1
	The pre-trained RNN, GRU and LSTM with meta-learner SVM	ArTwitter	For cross-validation, 92.24% of ACC, 92.39% of PRE, 92.24% of REC, 92.24% of F1
			For testing, 83.12% of ACC, 83.85% of PRE, 83.12% of REC, 83.06% of F1
	The pre-trained RNN, GRUand LSTM with meta-learner LR	AJGT	For cross-validation, 93.4% of ACC, 93.5% of PRE, 93.4% of REC, 93.4% of F1
			For testing, 86.11% of ACC, 86.13% of PRE, 86.11% of REC, F86.11% of F1

## Data Availability

All datasets used to support the findings of this study are available from the direct link in the dataset citations. In addition, the code will be available upon request.

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
