# Peer review of "Heterogeneous Ensemble Deep Learning Model for Enhanced Arabic Sentiment Analysis"

_sensors, 2022, doi:10.3390/s22103707_

Round 1

Reviewer 1 Report

The authors propose an optimized heterogeneous ensemble deep learning model for enhanced Arabic sentiment analysis. The paper has potential but it needs the following issues to be fixed before acceptance.
1. While explaining ensemble learning in their work, authors are required to explain the existing ensemble learning models proposed in the literature for the sentiment classification:
These papers also use ensemble learning for sentiment classification.

2. Why ensemble learning instead of a standalone model? Ensemble learning might not be computationally efficient always. Please justify those issues in the paper.
3. Please revise the literature review section to add limitations and gaps in existing works in a separate paragraph. 
4. Is it worth putting the optimized keyword in the title? I think it is not necessary there.
5. It is worth explaining the hybrid feature extraction method for the sentiment analysis in the paper as in the paper:

Reviewer 2 Report

In this work, the authors propose an optimized heterogeneous stacking ensemble model for enhancing the performance of Arabic sentiment analysis that combines three different pre-trained Deep Learning (DL) models (Recurrent Neural Network (RNN), Long Short-Term Memory (LSTM), Gated Recurrent Unit (GRU) in conjunction with three meta-learners Logistic Regression (LR), Random Forest (RF), Support Vector Machine (SVM) in order to enhance model’s performance for predicting Arabic sentiment analysis.

Please add the reference “MSA is commonly used in radio, newspapers, and television. All these reasons lead to the inflation and increase of Arab markets, and the expanding usage of media platforms Leads to the interest in analyzing data in Arabic despite its correctness and its multiplicity of dialects.”

Is the ethical approval obtained? If yes, please mention it in the method section.

Please add the references for the equations 3, 7-12, 13-16, 17-20,

The results are presented in a complex way. It would be better if the authors can present the results in a simpler version.

Please add the full form of abbreviations in the Figures and Tables. Abbreviations can be added in the figure captions and below the Tables.

Please present the conclusion in a simpler way.

Reviewer 3 Report

Congratulations !

Author Response

Comment 1:  Congratulations!

Authors’ response:

Thank you very much for your support.

Reviewer 4 Report

For improving the performance of Arabic sentiment analysis, this article suggested an optimized heterogeneous stacking ensemble model. To improve the model's performance for predicting Arabic sentiment analysis, the proposed model combines three different pre-trained Deep Learning (DL) models (Recurrent Neural Network (RNN), Long Short-Term Memory (LSTM), Gated Recurrent Unit (GRU) with three meta-learners (Logistic Regression (LR), Random Forest (RF), and Support Vector Machine (SVM). On three benchmark Arabic datasets, the performance of the proposed model with RNN, LSTM, GRU, and the five conventional ML techniques: Decision Tree (DT), LR, K-Nearest Neighbor (KNN), Random Forest (RF), and Naive Bayes (NB) is compared. Grid search and KerasTuner are used to optimize Machine Learning (ML) and Deep Learning (DL) parameters, respectively. To assess the models' performance and confirm the findings, we used accuracy, precision, recall, and the f1-measure. The obtained results have also been compared with the existing ones.  In summary, the paper contributes to the field and thus, I suggest its publication. 

Author Response

Comment 1:  In summary, the paper contributes to the field and thus, I suggest its publication. 

Authors’ response:

Thank you very much for your support.

Reviewer 5 Report

If possible, add the used literature sources and show them correctly in the text of the article.

Round 2

Reviewer 1 Report

The reviewer would like to thank the authors for their hard work to improve the manuscript. Given that the authors improved the manuscript significantly, the reviewer is inclined to accept the manuscript. 

Reviewer 5 Report

I find the authors quoting: Abdullah Alharbi source (34) from line 678 cited at line 134 and Shaker El-Sappagh sources: (22, 23 and 24) from lines: 643, 646 and 649 which is not cited anywhere.